# DeepTwist: Learning Model Compression via Occasional Weight Distortion

## Abstract

Model compression has been introduced to reduce the required hardware resources while maintaining the model accuracy. Lots of techniques for model compression, such as pruning, quantization, and low-rank approximation, have been suggested along with different inference implementation characteristics. Adopting model compression is, however, still challenging because the design complexity of model compression is rapidly increasing due to additional hyper-parameters and computation overhead in order to achieve a high compression ratio. In this paper, we propose a simple and efficient model compression framework called DeepTwist which distorts weights in an occasional manner without modifying the underlying training algorithms. The ideas of designing weight distortion functions are intuitive and straightforward given formats of compressed weights. We show that our proposed framework improves compression rate significantly for pruning, quantization, and low-rank approximation techniques while the efforts of additional retraining and/or hyper-parameter search are highly reduced. Regularization effects of DeepTwist are also reported.

## 1 Introduction

The number of parameters of deep neural networks (DNNs) is rapidly increasing in order to support various complex tasks associated with increasing amount of training data. For example, Deep Speech 2 has $10\times$ more parameters compared with the previous version (Narang et al., 2017). The Long Short-Term Memory (LSTM) (Hochreiter & Schmidhuber, 1997) model using PTB dataset (Marcus et al., 1993) requires exponential increase in number of parameters to achieve incremental improvement in test perplexity. Such a requirement of large DNN model size leads to not only long training time but also high latency and huge storage requirement for performing inference, creating a challenge for mobile applications to be deployed.

Note that the memory access dominates the entire inference energy if data reuse is not high enough, since computation units demand relatively low amount of energy (Han et al., 2016). Therefore, model compression is crucial to efficiently implementing DNN applications such as language models and speech recognition. As a result, model compression is actively being studied with the ideas such as low-rank matrix factorization (N. Sainath et al., 2013; Prabhavalkar et al., 2016), parameter pruning (Han et al., 2015), quantization (Xu et al., 2018; Zhu et al., 2017), knowledge distillation (Hinton et al., 2015; Polino et al., 2018), low precision (Zhou et al., 2016), and so on.

Model compression is based on Bayesian inference and prior information on the model parameters (Hinton & van Camp, 1993). Depending on specific implementation methods, various ways to exploit prior information exist (e.g., pruning and low-rank approximation). Up to now, model compression techniques have been independently developed to fully utilize the unique compression principles of each compression technique. Consequently, different model compression techniques demand dedicated hyper-parameters and compression-aware training methods. We observe in recent studies that the number of required additional hyper-parameters is increasing. Training algorithms are also being modified intensively to take into account the model compression effects in order to enhance the compression rate given a target accuracy. Such efforts, however, lead to significantly increased training time due to computation overhead and substantial hyper-parameter search space. Desirable compression techniques exhibit the following characteristics: 1) Existing training method is not modified because of model compression, 2) The addition to the number of hyper-parameters

exclusively dedicated to model compression is minimized, and 3) Nonetheless, high compression ratio should be achieved.

In this paper, we propose a unified model compression framework for performing a variety of model compression techniques. Our proposed framework, called DeepTwist, involves only one extra hyper-parameter regardless of the type of model compression while any modifications to the training algorithm is not necessary. For example, a pruning method based on DeepTwist does not need masking layers, which have often been entailed additionally to record individual pruning decision on each weight (Han et al., 2016; Guo et al., 2016). During the training process for model compression, weights are slightly and infrequently distorted in the intervals of training steps where weight distortion follows well-known basic model compression operations. In our proposed framework, weight distortion is considered as a weight-noise-injection process that has a potential to improve the accuracy. Because of no modifications required for the training algorithm and small computation overhead introduced from infrequent weight distortions, DeepTwist presents a practical and efficient model compression platform. We show that DeepTwist can provide high compression ratio for compression techniques including but not limited to weight pruning, quantization, and low-rank approximation.

## 2 RELATED WORKS

Parameter pruning is based on the observation that large number of parameters are redundant. Optimal brain damage finds redundant parameters using the second derivative of loss function at the cost of severe computation overhead (LeCun et al., 1990). Deep compression (Han et al., 2015) suggests a fast pruning method using the magnitude of parameters to determine the importance of each parameter. Dynamic network surgery (DNS) allows weight splicing to correct wrongly pruned parameters at the expense of additional hyper-parameters (Guo et al., 2016). Variational Dropout (Molchanov et al., 2017) introduces individual dropout rate for each parameter.

Quantization techniques widely used for digital signal processing are useful for DNNs as well. Researchers have suggested aggressive binary quantization techniques dedicated to DNNs, where computations are replaced with simple logic units. BinaryConnect (Courbariaux et al., 2015) learns quantized weights where the forward propagation is performed using quantized weights while full-precision weights are reserved for accumulating gradients. Scaling factors are stored additionally to compensate for the limited range of binary weights (Rastegari et al., 2016). While binary quantization achieves impressive amount of compression, its accuracy in large models (especially for RNNs) is degraded seriously (Xu et al., 2018). Thus, multi-bit quantization techniques to reduce mean squared error (MSE) between full-precision and quantized weights are introduced. For example, Alternating multi-bit quantization (Xu et al., 2018) extends the XNOR-Net architecture (Rastegari et al., 2016) to find a set of coefficients minimizing MSE, and recovers full precision accuracy with 3-4 bits per weight.

Researchers have also successfully compressed weight matrices using low-rank matrix factorization. A matrix is decomposed into two smaller matrices by reducing the rank through Singular Value Decomposition (SVD). Then, DNNs for acoustic modeling and language modeling can be reduced by 30-50% without noticeable accuracy degradation (N. Sainath et al., 2013). To further improve compression rate, two weight matrices inside an LSTM layer can share a projection layer to be commonly used for each decomposition (Prabhavalkar et al., 2016).

## 3 DEEPTWIST: LEARNING MODEL COMPRESSION FRAMEWORK

Our proposed model compression framework, DeepTwist, mainly performs full-precision training as if model compression is not being considered. Occasionally, however, weights are distorted as shown in Figure 1 such that the distorted weights follow the compressed form. For instance, in the case of weight pruning, some weights are converted into zeros as a weight distortion function. Distortion step, $S_D$, determines how frequently such distortions should be performed. As described in Figure 1, weight distortion at $N^{th}$ batch replaces the original weights with distorted weights to be used for training at $(N + 1)^{th}$ batch. For a successful model compression, DeepTwist is required to meet the following specifications:

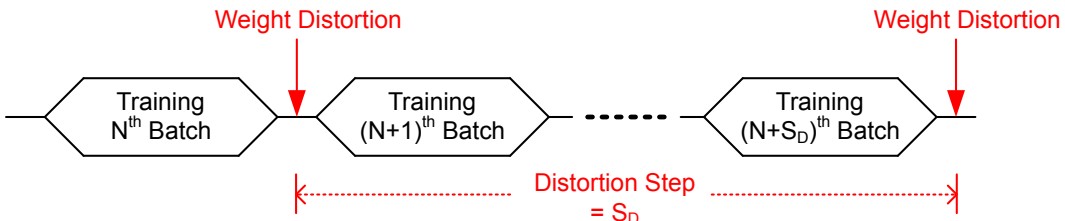

Figure 1: Training procedure for DeepTwist where weights are distorted occasionally.

- Full-precision training should converge and achieve the full accuracy in the end, even though weights are distorted at every $S_D$ step.

- The training should be finished right after the weights are distorted to hold the compression format of weights.

- Accuracy drop through a weight distortion step should decrease during training. In other words, the loss function degradation caused by a weight distortion should be gradually diminished.

From the above requirements of DeepTwist, one can notice that DeepTwist partly holds the characteristics of regularization techniques even though faster training or improving accuracy is not an immediate objective. The next section, indeed, shows that DeepTwist tends to enhance the accuracy via an appropriate amount of distortion.

Accuracy of compressed models highly depends on the flatness of the loss surface (Hochreiter & Schmidhuber, 1995). The motivation of DeepTwist is derived from the goal of reaching flatter minima based on careful addition of noise to weights (Hochreiter & Schmidhuber, 1995). Weight distortion can be regarded as a weight-noise-injection method which has been known to regularize models associated with flatter minima (Goodfellow et al., 2016). Flat minima reduce accuracy loss induced by model compression since DNNs become less sensitive to the variation of weights (Courbariaux et al., 2015; Hochreiter & Schmidhuber, 1995). For high-dimensional DNNs, it is known that most of the local minima have similar loss function values to that of the global optimum (LeCun et al., 2015). Nonetheless, local minima with similar DNN accuracy can represent vastly different amount of accuracy drop through compression depending on the flatness of the loss surface (Choromanska et al., 2015). Correspondingly, searching for the local minima exhibiting the minimum compression loss is the key to maximizing the compression ratio. DeepTwist enables locating such local minima via occasional weight distortions.

## 4 DEEPTWIST APPLICATIONS

In this section, we describe how DeepTwist can be applied in practice to implement model compression with various DNN models. We present the effect of varying $S_D$ on compression ratio and the weight distortion methods tailored for selected compression techniques, specifically, pruning, quantization, and low-rank approximation. Note that DeepTwist is versatile to function with other model compression methods as well (including even unknown ones yet) provided that a weight distortion function implies an inference implementation form of the compressed weights.

### 4.1 PRUNING

The initial attempt of pruning weights was to locate redundant weights by computing the Hessian to calculate the sensitivity of weights to the loss function (LeCun et al., 1990). However, such a technique has not been considered to be practical due to significant computation overhead for computing the Hessian. Magnitude-based pruning (Han et al., 2015) has become popular because one can quickly find redundant weights by simply measuring the magnitude of weights. Since then, numerous researchers have realized higher compression ratio largely by introducing Bayesian inference modeling of weights accompanying supplementary hyper-parameters.

For example, DNS permits weight splicing when a separately stored full-precision weight becomes larger than a certain threshold (Guo et al., 2016). Optimizing splicing threshold values, however, necessitates extensive search space exploration, and thus, longer training time. Variational dropout method (Molchanov et al., 2017) introduces an explicit Bayesian inference model for a prior distribution of weights, which also induces various hyper-parameters and increased computational complexity.

---

**Algorithm 1:** Weight distortion for pruning

---

**input** : Weight matrix $\boldsymbol{W}$, pruning rate $p$
**output:** Distorted weight matrix $\hat{\boldsymbol{W}}$
  1: $\hat{\boldsymbol{W}} \leftarrow \boldsymbol{W}$
  2: Sort the elements of $\hat{\boldsymbol{W}}$ by magnitude
  3: The $p\%$ smallest elements of $\hat{\boldsymbol{W}}$ (by magnitude) are set to zero

---

Algorithm 1 presents the distortion function for pruning weights using DeepTwist. Since DeepTwist does not need any additional layers, masking layers do not exist in our proposed pruning method. As a result, even though weights are pruned and replaced with zero at a distortion step, pruned weights are still updated in full precision until the next distortion step. If the amount of updates of a pruned weight grows large enough between two consecutive distortion steps, then the weight pruned at last distortion may not be pruned at the next distortion step. Such a feature (i.e., pruning decisions are not fixed) is also utilized for weight splicing in DNS (Guo et al., 2016). Weight splicing in DNS relies on a hysteresis function (demanding sophisticated fine-tuning process with associated hyper-parameters) to switch pruning decisions. Pruning decisions through DeepTwist, on the other hand, are newly determined at every $S_D$ step by Algorithm 1.

We present experimental results with LeNet-5 and LeNet-300-100 models on MNIST dataset which are also reported by Guo et al. (2016); Molchanov et al. (2017). LeNet-5 consists of 2 convolutional layers and 2 fully connected layers while 3 fully connected layers construct LeNet-300-100. We train both models for 20000 steps using Adam optimizer where batch size is 50. All the layers are pruned at the same time and the pruning rate increases gradually following the equation introduced in Zhu & Gupta (2017):

$$p_t = p_f + (p_i - p_f) \left( 1 - \frac{t - t_i}{t_f - t_i} \right)^E ,\qquad(1)$$

where $E$ is a constant, $p_f$ is the target pruning rate, $p_i$ is the initial pruning rate, $t$ is the current step, and the pruning starts at training step $t_i$ and reaches $p_f$ at training step $t_f$. After $t_f$ steps, pruning rate is maintained to be $p_f$. For LeNet-5 and LeNet-300-100, $t_i$, $p_i$, $E$ are 8000 (step), 25(%), and 7, respectively. $t_f$ is 12000 (step) for LeNet-5 and 13000 (step) for LeNet-300-100. Note that these choices are not highly sensitive to test accuracy as discussed in Zhu & Gupta (2017). We exclude dropout to improve the accuracy of LeNet-300-100 and LeNet-5 since pruning already works as a regularizer (Han et al., 2015; Wan et al., 2013). We keep the original learning schedule and the total number of training steps (no additional training time for model compression).

Table 1 presents the comparison on pruning rates and test accuracy. Despite the simplicity of our pruning technique, DeepTwist produces higher pruning rate compared with DNS and similar compared with variational dropout technique which involves much higher computational complexity. For Table 1, we use $S_D$=10 for LeNet-5 and $S_D$=5 for LeNet-300-100. We investigate how sensitive $S_D$ is to the test accuracy when the other parameters (such as pruning rates, learning rate, and total training time) are fixed. As shown in Table 2, for a wide range of $S_D$, the test accuracy has negligible fluctuation[1]. Too large $S_D$ would result in 1) too few weight distortion, 2) coarse-grained gradual pruning, and 3) unnecessarily large updates for correctly pruned weights. On the other hand, too small $S_D$ may yield excessive amount of weight distortion and reduce the opportunity for the pruned weights to recover.

We apply DeepTwist-based pruning to an RNN model to verify the effectiveness of DeepTwist with various models. We choose an LSTM model (Zaremba et al., 2014) on the PTB dataset (Marcus et al., 1993). Following the model structure given in Zaremba et al. (2014), our model consists of

---

[1]Even though we cannot show such a sensitivity study for all of the remaining experiments in this paper, $S_D$ has also shown low sensitivity to the accuracy even for other models and compression techniques.

Table 1: Pruning rate and accuracy comparison using LeNet-300-100 and LeNet-5 models on MNIST dataset. DC (Deep Compression) and Sparse VD represent a magnitude-based technique (Han et al., 2016) and variational dropout method (Molchanov et al., 2017), respectively.

| Model | Layer | Weight Size | Pruning Rate (%) | | | |
|-------|-------|-------------|------|------|-----------|-----------|
| | | | DC | DNC | Sparse VD | DeepTwist |
| LeNet-300-100 | FC1 | 235.2K | 92 | 98.2 | 98.9 | 98.9 |
| | FC2 | 30K | 91 | 98.2 | 97.2 | 96.0 |
| | FC3 | 1K | 74 | 94.5 | 62.0 | 62.0 |
| | Total | 266.2K | 92 | 98.2 | 98.6 | 98.4 |
| LeNet-5 | Conv1 | 0.5K | 34 | 85.8 | 67 | 60.0 |
| | Conv2 | 25K | 88 | 96.9 | 98 | 97.0 |
| | FC1 | 400K | 92 | 99.3 | 99.8 | 99.8 |
| | FC2 | 5K | 81 | 95.7 | 95 | 95.0 |
| | Total | 430.5K | 92 | 99.1 | 99.6 | 99.5 |

| Model | Accuracy (%) | | | |
|-------|------|------|-----------|-----------|
| | DC | DNC | Sparse VD | DeepTwist |
| LeNet-300-100 | 98.4 | 98.0 | 98.1 | 98.1 |
| LeNet-5 | 99.2 | 99.1 | 99.2 | 99.1 |

Table 2: Test accuracy (average of 10 runs for the choice of each $S_D$) when pruning weights of LeNet-5 model using DeepTwist. Pruning rates are described in Table 1.

| Distortion Step($S_D$) | 1 | 2 | 5 | 10 | 50 | 100 | 200 | 500 |
|------------------------|-----|-----|-----|-----|-----|-----|-----|-----|
| Accuracy(%) | 99.00 | 99.06 | 99.06 | 99.11 | 99.05 | 98.98 | 98.72 | 96.52 |

Table 3: Comparison on perplexity using various pruning rates. $p_f$ is the target pruning rates for the embedded layer, LSTM layer, and softmax layer.

| Model Size | Pruning Method | Perplexity | | | | | | |
|------------|----------------|--------|------|------|------|------|------|-------|
| | | $p_f=$ | 0% | 80% | 85% | 90% | 95% | 97.5% |
| Medium | (Zhu & Gupta, 2017) | | 83.37 | 83.87 | 85.17 | 87.86 | 96.30 | 113.6 |
| (19.8M) | DeepTwist | | 83.78 | 81.54 | 82.62 | 84.64 | 93.39 | 110.4 |
| Large | (Zhu & Gupta, 2017) | | 78.45 | 77.52 | 78.31 | 80.24 | 87.83 | 103.20 |
| (66M) | DeepTwist | | 78.07 | 77.39 | 77.73 | 78.28 | 84.69 | 99.69 |

an embedding layer, 2 LSTM layers, and a softmax layer. The number of LSTM units in a layer can be 200, 650, or 1500, depending on the model configurations (referred as small, medium, and large model, respectively). The accuracy is measured by Perplexity Per Word (PPW), denoted simply by perplexity in this paper. We apply gradual pruning with $E = 3$, $t_i = 0$, $p_i = 0$, $t_f = 3^{rd}$ epoch (for medium) or $5^{th}$ epoch (for large) to the pre-trained PTB models. DeepTwist pruning for the PTB models is performed using $S_D = 100$ and the initial learning rate is 2.0 for the medium model (1.0 for pre-training) and 1.0 for the large model (1.0 for pre-training) while the learning policy remains to be the same as in Zaremba et al. (2014).

For all of the pruning rates selected, Table 3 shows that DeepTwist improves perplexity better than the technique in Zhu & Gupta (2017) which is based on Han et al. (2015). Interestingly, when the pruning rate is 80% or 85%, perplexity of the retrained model is even smaller than that of the pre-trained model because of the regularization effect of DeepTwist. The superiority of DeepTwist-based pruning is partly supported by the observation that non-zero weights successfully avoid to be small through retraining while the conventional pruning still keeps near-zero (unmasked) weights (see Appendix for weight distribution difference after retraining).

## 4.2 QUANTIZATION

Following the Binary-Weight-Networks (Rastegari et al., 2016), a weight vector $\boldsymbol{w}$ is approximated to be $\alpha\boldsymbol{b}$ by using a scaling factor $\alpha$ and a vector $\boldsymbol{b} (= \{-1, +1\}^n)$, where $n$ is the vector size. Then $||\boldsymbol{w} - \alpha\boldsymbol{b}||^2$ is minimized to obtain

$$\boldsymbol{b}^* = \text{sign}(\boldsymbol{w}), \ \alpha^* = \frac{\boldsymbol{w}^\top \boldsymbol{b}^*}{n}. \tag{2}$$

1-bit quantization shown in Eq. (2) is extended to multi-bit ($k$-bit) quantization using a greedy method (Guo et al., 2017) where the $i^{\text{th}}$-bit ($i > 1$) quantization is performed by minimizing the residue of $(i-1)^{\text{th}}$-bit quantization as following:

$$\min_{\alpha_i, \boldsymbol{b}_i} ||\boldsymbol{r}_{i-1} - \alpha_i \boldsymbol{b}_i||^2, \ \text{where} \ \boldsymbol{r}_{i-1} = \boldsymbol{w} - \sum_{j=1}^{i-1} \alpha_j \boldsymbol{b}_j, \ 1 < i \le k. \tag{3}$$

The optimal solution of Eq. (3) is given as

$$\boldsymbol{b}_i^* = \text{sign}(\boldsymbol{r}_{i-1}), \ \alpha_i^* = \frac{\boldsymbol{r}_{i-1}^\top \boldsymbol{b}_i^*}{n}. \tag{4}$$

Note that Eq. (4) is not the optimal solution for $||\boldsymbol{w} - \sum_{i=1}^{k} \alpha_i \boldsymbol{b}_i||$. As an attempt to lower quantization error, $\{\alpha_i\}_{i=1}^{k}$ can be refined as $[\alpha_1, ..., \alpha_k] = \left( \left( \boldsymbol{B}_k^\top \boldsymbol{B}_k \right)^{-1} \boldsymbol{B}_k^\top \boldsymbol{w} \right)^\top$, when $\boldsymbol{B}_k = [\boldsymbol{b}_1, ...\boldsymbol{b}_k]$ (Guo et al., 2017). Further improvement can be obtained by using Alternating multi-bit method (Xu et al., 2018), where $\boldsymbol{B}_k$ is obtained by binary search given a new refined $\{\alpha_i\}_{i=1}^{k}$, and $\{\alpha_i\}_{i=1}^{k}$ and $\boldsymbol{B}_k$ are refined alternatively. This procedure is repeated until there is no noticeable improvement in MSE. Quantized weights are used and updated during training procedures associated with special considerations on quantized weights such as weight clipping and "straight-through estimate" (Xu et al., 2018).

---

**Algorithm 2:** Weight distortion for quantization

**input** : Weight matrix $\boldsymbol{W}$, number of quantization bits $k$
**output:** Distorted weight matrix $\hat{\boldsymbol{W}}$
  1: $\hat{\boldsymbol{W}} \leftarrow \boldsymbol{W}$
  2: Apply a quantization method (such as greedy method or Alternating method) to $\hat{\boldsymbol{W}}$
  3: Convert the elements of $\hat{\boldsymbol{W}}$ to full-precision values (i.e., dequantization)

---

Recently, researchers have proposed 1-bit quantization techniques reaching full precision accuracy of convolutional neural networks (CNNs). RNNs, however, still have a difficulty to be quantized with a low number of quantization bits. Hence, we apply DeepTwist-based quantization using the distortion function described in Algorithm 2 to an LSTM model with one hidden layer of 300 units on the PTB dataset, which is also demonstrated in Xu et al. (2018); He et al. (2016); Hubara et al. (2016). The weights of the embedding layer, LSTM layer, and softmax layer are quantized by using 1-3 bits.

DeepTwist-based quantization experiments with a pre-trained model are conducted by following the settings in Xu et al. (2018). We, however, do not clip the weights for retraining because weights do not explode with DeepTwist and the accuracy is improved faster without weight clipping. For weight distortion, we utilize a greedy method and Alternating method. The number of epochs for retraining is equivalent to that of pre-training. Table 4 presents the quantization comparison with the same model structures although the choice of hyper-parameters (such as learning method, initial learning rate, and so on) can be different. Note that we do not quantize activations since 1) quantizing on-the-fly slows down the inference operations with additional hardware resource requirements, 2) quantizing activation does not reduce the external memory footprint and accesses, and 3) RNNs are memory-intensive models such that the gain of simple internal logic operations derived by quantized activations is low (Narang et al., 2017). Using DeepTwist with simple and quick quantization method, we accomplish test perplexity comparable with Alternating quantization method (Xu et al., 2018), which requires costly quantization process (e.g., iterative least sqaures solution and binary search) at every step.

Table 4: Perplexity comparison using various quantization methods with a hidden LSTM layer of size 300 on the PTB dataset. For DeepTwist, $S_D$=2000 and the initial learning rate for retraining is 20.0. The function used for weight distortion ($f_D$) is described in comments.

| Quantization Technique | Weight Bits | Activation Bits | Perplexity | Comment |
|---|---|---|---|---|
| QNNs (Hubara et al., 2016) | 32 | 32 | 97 | Reference |
| | 2 | 3 | 220 | |
| | 3 | 4 | 110 | |
| | 4 | 4 | 100 | |
| Balanced Quantization (He et al., 2016) | 32 | 32 | 109 | Reference |
| | 1 | 32 | 198 | |
| | 2 | 3 | 142 | |
| | 3 | 3 | 120 | |
| | 4 | 4 | 114 | |
| Alternating Quantization (Xu et al., 2018) | 32 | 32 | 89.8 | Reference |
| | 2 | 2 | 95.8 | |
| | 2 | 3 | 91.9 | |
| | 3 | 3 | 87.9 | |
| DeepTwist Quantization | 32 | 32 | 89.3 | Reference |
| | 1 | 32 | 110.2 | $f_D$: Binary Method [a] |
| | 2 | 32 | 93.7 | $f_D$: Greedy Method |
| | 2 | 32 | 91.7 | $f_D$: Alternating Method |
| | 3 | 32 | 89.0 | $f_D$: Greedy Method |
| | 3 | 32 | 86.7 | $f_D$: Alternating Method |

[a] For a binary quantization, greedy method and Alternating method are equivalent.

### 4.3 LOW-RANK APPROXIMATION

Low-rank approximation minimizes a cost function that measures how close an approximated matrix is to the original matrix subject to a reduced rank. Singular-value decomposition (SVD) minimizes the Frobenius norm of the matrix difference between the original one and the approximated one. SVD-based compression techniques have demonstrated that reduced rank can preserve the model accuracy for a variety of deep learning tasks including speech recognition and language modeling (N. Sainath et al., 2013; Prabhavalkar et al., 2016). Unlike pruning and quantization that demand special hardware considerations to fully exploit the sparsity (Lee et al., 2018) or bit-level operations (Wu et al., 2018), SVD is inherently a structural compression technique.

Each LSTM layer contains inter-layer weight matrix ($\boldsymbol{W}_x$) and recurrent weight matrix ($\boldsymbol{W}_h$). Instead of performing SVD (in the form of $\boldsymbol{U}\Sigma\boldsymbol{V}^\top$) on $\boldsymbol{W}_x$ and $\boldsymbol{W}_h$ separately, it is possible to obtain $\boldsymbol{V}_h^\top$ of $\boldsymbol{W}_h$ with reduced rank $r$ first, and then $\boldsymbol{W}_x = \boldsymbol{Z}_x\boldsymbol{V}_h^\top$ where $\boldsymbol{Z}_x$ is computed by the following:

$$\boldsymbol{Z}_x = \underset{\boldsymbol{Y}}{\operatorname{argmin}} ||\boldsymbol{Y}\boldsymbol{V}_h^\top - \boldsymbol{W}_x||_{\mathcal{F}}^2, \tag{5}$$

where $||\boldsymbol{X}||_{\mathcal{F}}$ denotes the Frobenius norm of the matrix $\boldsymbol{X}$. Since $\boldsymbol{V}_h^\top$ (a projection layer) is shared by $\boldsymbol{W}_x$ and $\boldsymbol{W}_h$, the compression ratio is improved while the accuracy drop may not be noticeable for DNN applications (Prabhavalkar et al., 2016).

---

**Algorithm 3:** Weight distortion for low-rank approximation

---

**input** : Weight matrix $\boldsymbol{W} \in \mathbb{R}^{m \times n}$, reduced rank $r$
**output:** Distorted weight matrix $\hat{\boldsymbol{W}} \in \mathbb{R}^{m \times n}$
  1: Decompose $\boldsymbol{W}$ into $\boldsymbol{U}\Sigma\boldsymbol{V}^\top$ through SVD where $\Sigma$ is an $m \times n$ diagonal matrix
  2: Obtain $\hat{\Sigma}$ by replacing all the singular values of $\Sigma$ after $r^{th}$ singular value with zeros
  3: $\hat{\boldsymbol{W}} \leftarrow \boldsymbol{U}\hat{\Sigma}\boldsymbol{V}^\top$

---

Let $\boldsymbol{W} \in \mathbb{R}^{m \times n}$ be a weight matrix with $m > n$. Weight distortion function for DeepTwist-based low-rank approximation is given as Algorithm 3. Note that the structures of $\hat{\boldsymbol{W}}$ and $\boldsymbol{W}$ are the same

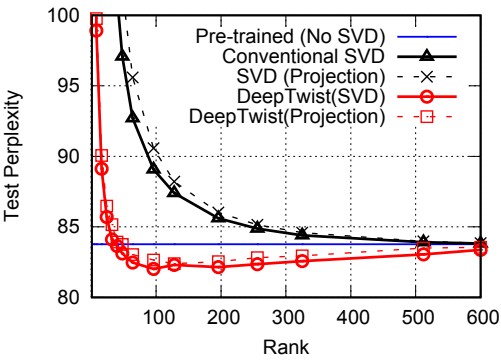 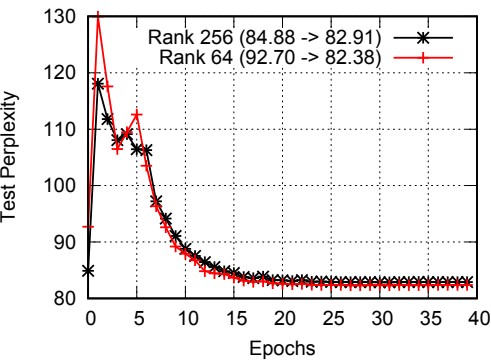

(a) Text perplexity with different rank $r$ using the con-
ventional SVD and DeepTwist-based SVD. Projection
layer incurs slight increase in perplexity.

(b) Test perplexity transitions with distorted weights
sampled once at each epoch. The test perplexity by the
first SVD without DeepTwist is 92.70 and 84.88 when
$r$=64 and 256, respectively.

Figure 2: Test perplexity using DeepTwist-based SVD on the PTB dataset.

during training. After finishing training, we truncate corresponding rows and/or columns of $\boldsymbol{U}$, $\hat{\Sigma}$, and $\boldsymbol{V}^\top$ after $r^{th}$ singular value. Then, $\hat{\boldsymbol{W}}$ is divided into an $m \times r$ matrix (from truncated $\boldsymbol{U}\hat{\Sigma}$) and an $r \times n$ matrix (from truncated $\boldsymbol{V}^\top$). The total number of elements is then $r(m + n)$ which can be much smaller than $mn$ if $r \ll m, n$. The compression ratio with an SVD becomes $mn/(r(m + n))$. DeepTwist-based SVD compression reduces the rank $r$ until the compression ratio is maximized while we maintain the accuracy of full-rank weight matrices.

DeepTwist-based SVD compression is applied to pre-trained medium and large PTB models previously used for our pruning experiments. For the medium PTB model, $S_D$=100 and the initial learning rate is fixed to be 2.6 (1.0 for pre-training) regardless of reduced rank $r$. The number of epochs for retraining is the same as that for pre-training. Figure 2a describes the test perplexity of the conventional SVD method and DeepTwist-based method when $r$ is reduced from the maximum 650 (LSTM size). Figure 2a considers a projection layer as well. As shown in Figure 2a, DeepTwist provides improved perplexity compared with the conventional SVD for the entire range of reduced rank $r$. Surprisingly, text perplexity achieved by DeepTwist-based SVD can be much lower than that of pre-trained model(=83.78) even with significantly reduced rank[2]. Note that even though Figure 2a and 2b are produced by using one fixed initial learning rate, the optimal initial learning rate achieving the best perplexity is different for each reduced rank $r$. For instance, given $r$=512, 256, and 128, we obtain the test perplexity of 82.62, 81.75, and 81.70, respectively (lower than in Figure 2a), when the initial learning rates are 1.8, 2.0, and 2.2, respectively. In general, higher $r$ requires a lower initial learning rate to improve the perplexity.

As shown in Figure 2b, test perplexity increases temporarily on the early epochs of retraining, and then, decreases gradually. The amount of retraining time to reach the minimum perplexity is about half the pre-training time. To examine the way DeepTwist explores search space to find suitable local minima for SVD, Figure 3 presents singular-value spectrum using $\Sigma$ of (non-distorted) $\boldsymbol{W}_x$ and $\boldsymbol{W}_h$ of layer 1 (layer 2 presents similar results). We can observe that through retrainig using DeepTwist, the magnitude of singular values grows before $r^{th}$ and shrinks after $r^{th}$.

The initial learning rate of the medium PTB model for DeepTwist-based SVD is chosen to be higher than the initial learning rate for pre-training (1.0). We conjecture that this high initial learning rate is allowed and preferred due to a strong regularization effect caused by DeepTwist with SVD. The large PTB model is, however, well compressed by DeepTwist with a low initial learning rate (associated with less amount of perplexity fluctuations compared with Figure 2b). See Appendix for the details of SVD results on the large PTB model. It would be necessary to study the relationship between the model configurations and the required amount of regularization effects by DeepTwist.

---

[2]For instance in Figure 2a, when $r$=96, test perplexity with DeepTwist becomes 82.02 (without a projection layer) or 82.65 (with a projection layer), both are lower than 83.78 (of the pre-trained model).

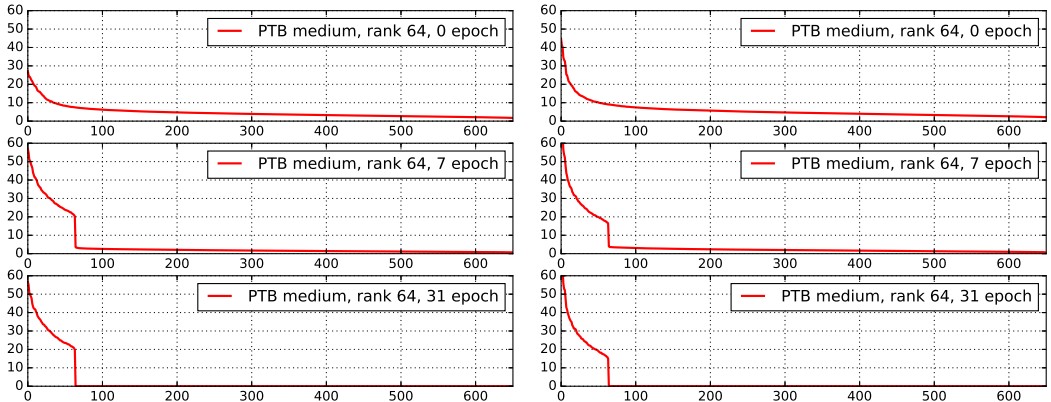

Figure 3: Singular-value spectrum of the diagonal (non-distorted) matrix $\Sigma$ sampled from the pre-trained model (0 epoch), at the $7^{th}$ epoch, and at the $31^{st}$ epoch with $r$=64. (Left): from $\boldsymbol{W}_x$ of layer 1, (Right): from $\boldsymbol{W}_h$ of layer 1.

## 5 CONCLUSION

In this paper, we proposed a general model compression framework called DeepTwist. DeepTwist does not modify the training algorithm for model compression, but introduces occasional weight distortions. We demonstrated that existing model compression methods (pruning, quantization, and low-rank approximation) can be improved by DeepTwist. DeepTwist provides a platform where new compression techniques can be suggested as long as a weight distortion function is designed along with a corresponding compression implementation form. Since DeepTwist is a weight-noise-injection process, the model accuracy after model compression can be even better than that of the pre-trained model. As future works, it would be interesting to study the convergence mechanism and regularization effect of DeepTwist, and apply DeepTwist to other compression methods such as low-precision (of fixed point) technique.

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

# A APPENDIX

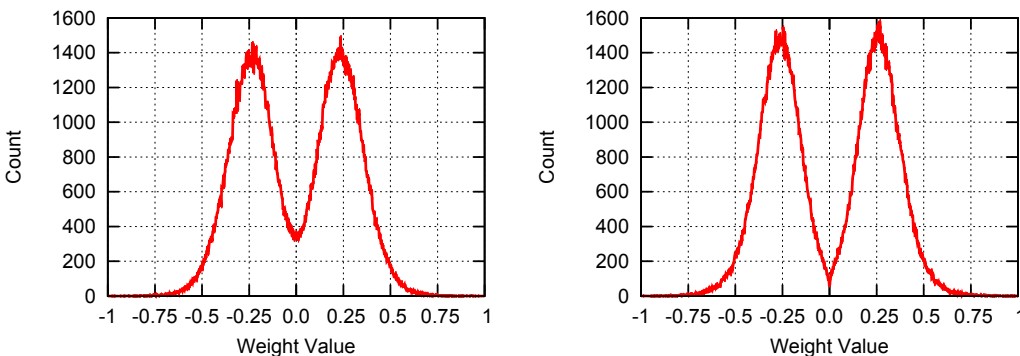

Figure 4: Weight distribution of LSTM layer 1 of the medium PTB model after retraining with (Left) a magnitude-based pruning and (Right) DeepTwist-based pruning with 90% pruning rate. DeepTwist incurs a sharp drop in the count of near-zero weights.

Table 5: Test perplexity of the large PTB model after DeepTwist-basd SVD compression using various initial learning rates for retraining (without a projection layer). Test perplexity after the first SVD compression (which is the test perplexity by using the conventional SVD) is shown on the left side. We train the model for 55 epochs for both pre-training and retraining.

| Rank | SVD | Initial Learning Rate for Retraining | | | | |
| --- | --- | --- | --- | --- | --- | --- |
| | | 0.6 | 0.8 | 1.0 | 1.2 | 1.4 |
| 600 | **79.251** | 78.166 | **78.130** | 78.716 | 79.561 | 80.707 |
| 512 | **79.646** | 77.875 | **78.049** | 78.367 | 79.164 | 79.646 |
| 256 | **83.241** | 77.504 | **77.409** | 77.656 | 77.964 | 78.723 |
| 192 | **86.053** | 77.525 | **77.495** | 77.497 | 77.528 | 78.367 |
| 128 | **93.722** | 77.711 | 77.694 | **77.611** | 77.738 | 78.172 |
| 96 | **104.078** | 78.144 | 77.961 | **77.840** | 78.019 | 78.812 |
| 64 | **130.946** | 79.074 | **78.718** | 78.768 | 78.741 | 79.418 |
| 32 | **381.992** | 81.581 | 81.283 | **81.169** | 81.350 | 81.982 |

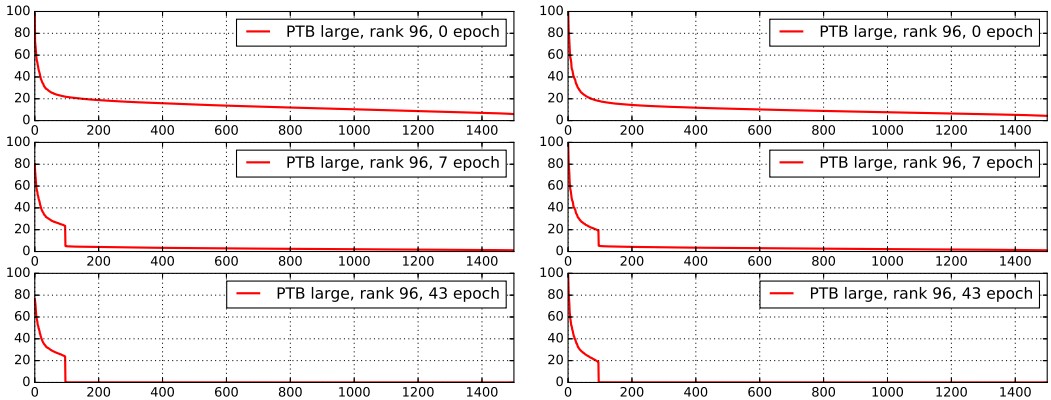

Figure 5: SV spectrum for the PTB large model when $r = 96$. (Left): from the (non-distorted) inter-layer weights of layer 1, (Right): from the (non-distorted)recurrent weights of layer 1. Layer 2 presents similar results. The number of hidden LSTM units is 1500.

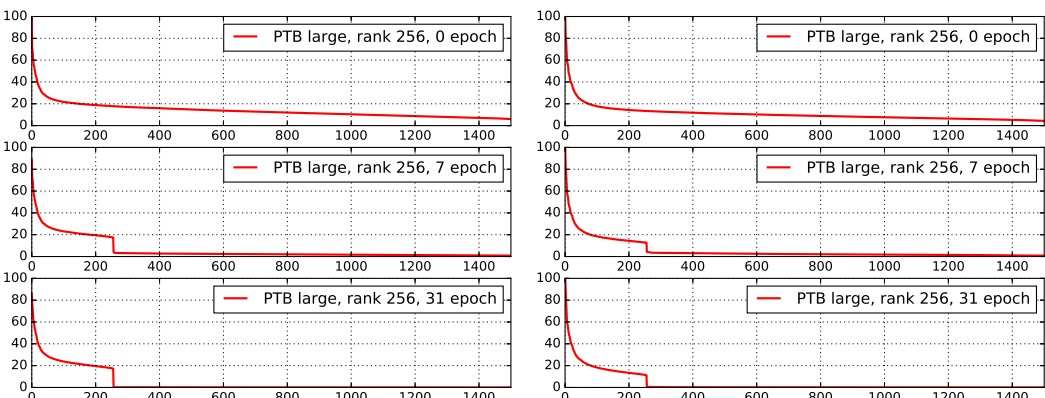

Figure 6: SV spectrum for the PTB large model when $r = 256$. (Left): from the (non-distorted) inter-layer weights of layer 1, (Right): from the (non-distorted) recurrent weights of layer 1. Layer 2 presents similar results. The number of hidden LSTM units is 1500.

