# OpenReview forum: "DeepTwist: Learning Model Compression via Occasional Weight Distortion"
_ICLR.cc/2019/Conference_

### Official Review · AnonReviewer3 · 2018-10-31
**limited novelty**

**Rating:** 4
**Confidence:** 4

**Review:**

This paper proposed a general framework, DeepTwist, for model compression. The so-called weight distortion procedure is added into the training every several epochs. Three applications are shown to demonstrate the usage of the proposed approach.

Overall, I think the novelty of the paper is very limited, as all the weight distortion algorithms in the paper can be formulated as the proximal function in proximal gradient descent. See http://www.stat.cmu.edu/~ryantibs/convexopt-S15/scribes/08-prox-grad-scribed.pdf for a reference.

Specifically, the proposed framework can be easily reformulated as a loss function plus a regularizer for proximal gradient. Using gradient descent (GD), there will be two steps: (1) finding a new solution using GD, and (2) project the new solution using proximal function. Now in deep learning, since SGD is used for optimization, several steps are need to locate reasonable solutions, i.e. the Distortion Step in the framework. Then proximal function can be applied directly after Distortion Step to project the solutions. In this way, we can easily see that the proposed framework is a stochastic version of proximal gradient descent. Since SGD is used for training, several minibatches are needed to achieve a relatively stable solution for projection using the proximal function, which is exactly the proposed framework in Fig. 1.

PS: After discussion, I think the motivation of the method is not clear to understand why the proposed method works.

---

> ### Author Response · Authors · 2018-11-10
> **Response to AnonReviewer3**
>
> Thank you for the review.
>
> While formulating a proximal function for model compression might be an interesting idea (if search space is highly limited) as the reviewer suggested, we believe that our proposed method is fundamentally different from proximal gradient descent approaches due to the following reasons:
>
> 1) Proximal gradient descent is meant to solve a convex optimization problem while our aim is to solve a non-convex problem in which each local minimum exhibits vastly different test accuracy after compression. Jumping to another local minimum from a certain minimum would not be easily achieved by convex optimization methods.
> 2) Finding a particular flat minimum is the key to obtaining good model compression (and good generalization as well). Such an exploration, however, cannot be obtained by a proximal function since we need to investigate lots of different local minima with different amount of flatness in loss surface.
> 3) While proximal gradient descent can be useful to find a certain local minimum close to the starting point given a convex constraint, wide exploration (associated with possibly transient accuracy loss in the initial training as shown in Figure 2.(b)) is necessary to escape from a point with sharp loss surface.  Investigating many different local minima would be only available with large learning rate (as we have chosen for our experiments) and/or large amount of weight distortion.
> 4) Our effort to introduce optimal distortion step size and learning rate for a given compression problems is connected to exploration, not exploitation (which potentially supported by proximal functions where convergence matters).
>
> Even though proximal gradient descent selects step size only considering convergence, Figure 1 can lead to the results such as Figure 2(b) which cannot be obtained if only local exploitation is employed.
>
> Finding a flat minimum has been known to be a difficult work as shown in the paper “On large-batch training for deep learning: generalization gap and sharp minima”, ICLR 2016. We firmly believe that our search space exploration method based on optimal distortion step size and amount of weight distortion enable us to produce better local minima well-suited to various model compression techniques.
> In short, unfortunately, we have failed to understand how you could connect our technique to proximal functions and proximal gradient descent.
> We strongly hope that you reconsider your decision.

---

> > ### Comment · AnonReviewer3 · 2018-11-10
> > **Thanks for the reply, but I will keep the same rate.**
> >
> > Below are my answers:
> >
> > 1. Proximal gradient descent can be used for nonconvex optimization. Just google it and you will find many papers on this topic.
> >
> > 2. Let me clarify my review.
> >
> > The proposed framework can be easily reformulated as a loss function plus a regularizer for proximal gradient. Using gradient descent (GD), there will be two steps: (1) finding a new solution using GD, and (2) project the new solution using proximal function.
> >
> > Now in deep learning, since SGD is used for optimization, several steps are need to locate reasonable solutions, i.e. the Distortion Step in the framework. Then proximal function can be applied directly after Distortion Step to project the solutions.
> >
> > In this way, we can easily see that the proposed framework is a stochastic version of proximal gradient descent.

---

> > > ### Author Response · Authors · 2018-11-10
> > > **Why low novelty?**
> > >
> > > Thank you so much for rapid response and kind clarification.
> > >
> > > We see your point and agree that our technique can follow the form of proximal function.
> > > But overall, we are wondering why the fact that DeepTwist can be formulated as a proximal function should be considered as limited novelty.
> > > Do you have concerns on the numbers of compression rate or accuracy we have shown in the paper?
> > > Or is there a similar work already published previously?
> > > We would greatly appreciate if you can explain why our work shows low novelty if our technique can be explained as proximal gradient descent.

---

> > > > ### Comment · AnonReviewer3 · 2018-11-10
> > > > **GD vs SGD**
> > > >
> > > > The reason I think this work is of limited novelty is due to the *stochastic* version of proximal gradient descent. This is just another way of solving the optimization problem, which is straightforward.

---

> > > > > ### Author Response · Authors · 2018-11-11
> > > > > **Performing model compression based on proximal gradient descent could be of great novelty**
> > > > >
> > > > > As we discuss in the paper, previous model compressions have been proposed in a heuristic way while each model compression technique has been developed separately and independently. If there is a straightforward and general ‘optimization’ method which can support different model compression techniques, we believe it should be considered to be of high novelty as the first attempt to solve various model compression issues using a general framework, such as ‘stochastic’ proximal gradient descent or DeepTwist. Since we present such a general optimization method for model compression for the first time associated with state-of-the-art compression rate, your suggestion that our technique can be explained with proximal gradient descent could be a strong claim that our technique solves critical problems in model compression (i.e., heuristic method dedicated to each technique with lots of additional hyper-parameters which cannot be formulated by using proximal function forms).
> > > > >
> > > > > 1. It has been challenging to find a good prior probability distribution on weights for a particular model compression format. Hence, instead of searching for a regularization form, lots of heuristic ways to obtain prior information on weights have been developed. For example, weight pruning is mainly performed by their magnitude (due to practical computation issues) instead of Hessian computation which can lead to higher compression rate. Our work has shown that a systematic way to perform ‘optimization’ for model compression without such heuristics is possible.
> > > > >
> > > > > 2. All the numbers of accuracy in our paper are obtained after weight distortion, not during normal training with full-precision weights. Our work is the first one that the accuracy after model compression can be achieved by a general framework (such as proximal function), unlike previous model compression which cannot be achieved by a simple and straightforward optimization (instead, all previous works modify training algorithms significantly, like adopting masking layers, temporal full-precision weights for quantization, feedforward and backward paths with different weights, and so on).
> > > > >
> > > > > 3. To present more details, for low-rank approximation, retraining after SVD to enhance accuracy has been a great challenge, hence no proximal functions have existed in the literature. Masking layers which have been essential for pruning also cannot be accommodated by using proximal functions. Our work is the first one in which masking layers are not necessary.
> > > > >
> > > > > 4. We are the first one that even if model compression is optimized by proximal functions, the accuracy still convergences under the constraints for model compression. To the best of our knowledge, there is no any previous papers showing that model compression can be done with a proximal function (or its variants).
> > > > >
> > > > > In the future, it would be of great importance to further develop and analyze model compression techniques using the knowledge on proximal gradient descent. We believe your suggestion is precious since it is a strong support that our technique is not only useful for model compression but also useful for suggesting a new regularization method based on various model compression formats.

---

> > > > > > ### Comment · AnonReviewer3 · 2018-11-14
> > > > > > **proximal gradient descent is *my* understanding/hypothesis**
> > > > > >
> > > > > > First of all, we need to be clear that interpreting the proposed approach using proximal gradient descent is *my* understanding/hypothesis, NOT THE CONTRIBUTION OF THIS SUBMISSION AT ALL, even though the authors agreed with it. If the proposed approach was formulated in a proximal gradient descent way and justified by theoretical or empirical evidence, I might raise my rate because this would be very useful to explain why the approach worked in different problems. Unfortunately, the paper in current shape does not meet it.
> > > > > >
> > > > > > Secondly, Reviewer 2 has similar questions on the importance of the approach.
> > > > > >
> > > > > > In summary, I will keep my rate unchanged at this point.

---

> > > > > > > ### Author Response · Authors · 2018-11-14
> > > > > > > **Our response**
> > > > > > >
> > > > > > > Thanks for the response and your patience.
> > > > > > >
> > > > > > > We absolutely acknowledge that such interpreting is your understanding and we have no any intention to claim anything regarding proximal gradient approximation.
> > > > > > >
> > > > > > > But please understand that because the first comment discusses proximal gradient descent as your major concern, we could not avoid connecting your understanding to the contributions of this paper.
> > > > > > > Also please refer to our response above (newly attached) regarding the list of contributions.

---

### Official Review · AnonReviewer1 · 2018-11-02
**The significance of the proposed method is limited**

**Rating:** 4
**Confidence:** 3

**Review:**

A model compression framework, DeepTwist, was proposed which makes the weights zero if they are small in magnitude. They used different model compression techniques in this framework to show the effectiveness of the proposed method.

This paper proposes a framework intending to use fewer hardware resources without compromising the model accuracy. However, when the weights are set to zero the weight matrix became sparser but still requires the whole weight matrix to be used by the computing resources, as removing some of the weights based on the sorting will not remove a node, only removes some of the connection with that node. Therefore, it is not clear how the proposed framework is helping the model compression techniques.

---

> ### Author Response · Authors · 2018-11-10
> **Response to AnonReviewer1**
>
> Thank you for the review.
>
> First, we want to mention that DeepTwist is proposed not only for weight pruning, but also for other compression techniques, such as quantization and low-rank approximation, as we discussed in Section 4.2 and 4.3
>
> After weight pruning is performed and zero weights are removed, we usually obtain a sparse matrix to represent non-zero weights. There are lots of existing sparse matrix computation libraries to support SpMV (sparse matrix-vector multiplication) and so on. If a matrix is highly sparse, then we would reduce memory footprint and amount of computations (for example, we can skip zero weights during computation) significantly.
> There have been extensive studies of efficient hardware implementation after weight pruning, and we want you to refer to the paper “EIE: efficient inference engine on compressed deep neural network” or “Deep compression: compressing deep neural networks with pruning, trained quatization and Huffman coding.”
> In this paper, we have not discussed particular sparse matrix implementation methods which are not our focus in this paper.
>
> We would greatly appreciate if you can reconsider your decision based on our comments and other methods we also discussed (i.e., quantization and low-rank approximation).

---

> > ### Comment · AnonReviewer1 · 2018-12-09
> > **Non-structurally sparsified models**
> >
> > Applying the proposed framework for weight pruning, quantization and low-rank approximation results in a non-structurally sparsified models which would not help significantly in speedup. I would refer the authors to the paper “Learning Structured Sparsity in Deep Neural Networks” for more details. I will keep my initial decision unchanged.

---

> > > ### Author Response · Authors · 2018-12-10
> > > **Our response**
> > >
> > > Thank you for your comment.
> > >
> > > Please understand that implementation issues after model compression is out of the scope in this paper.
> > > An analysis on the speed-up would need to discuss current hardware design/architecture issues.
> > > We do not suggest how much speed-up would be obtained by our techniques because there are lots of attempts to address this speed-up issue separately and differently in various perspectives (i.e., architecture, algorithm, novel devices, and so on).
> > >
> > > Fine-grained pruning inference can be expedited by various methods.
> > > (e.g., 'Viterbi-based pruning for sparse matrix with fixed and high index compression ratio' and 'Double Viterbi: weight encoding for high compression ratio and fast on-chip reconstruction for deep neural network')
> > > Low-rank approximation is inherently a structural compression method since the forms after compression still follow matrix multiplications.
> > > Quantization is also a structural compression while bit-level manipulation can be best performed by ASICs or FPGAs.
> > >
> > > We hope that you consider your rate again based on the computational advantage, compression rate, and accuracy in this paper.

---

> > > > ### Author Response · Authors · 2018-12-10
> > > > **One more comment**
> > > >
> > > >
> > > > Let us clarify that we have not mentioned that DeepTwist cannot support a structurally sparsified model in the manuscript
> > > >
> > > > Section 4 presents just a few examples of DeepTwist-based techniques and those 3 examples in Section 4 should be considered only as a subset of models that DeepTwist can support.
> > > > We would test more various models using DeepTwist as we described in the conclusion as the future work.

---

### Official Review · AnonReviewer2 · 2018-11-03
**A simple repeated compress and fine-tune method.**

**Rating:** 5
**Confidence:** 3

**Review:**

The paper does not really propose a new way of compressing the model weights, but rather a way of applying existing weight compression techniques. Specifically, the proposed solution is to repeatedly apply weight compression and fine-tuning over the entire training process. Unlike the existing work, weight compression is applied as a form of weight distortion, i.e. the model has the full degree of freedom during fine-tuning (to recover potential compression errors).

Pros:

- The proposed method is shown to work with existing methods like weight pruning, low-rank compression and quantization.


Cons:

- The idea is a simple extension of existing work.
- In Table 4, it is hard to compare DeepTwist with the other methods because activation quantization is not used.

---

> ### Author Response · Authors · 2018-11-10
> **Response to AnonReviewer2**
>
> Thank you for the review.
>
> While the weight formats after model compression follow well known ones, our model compression method is significantly different from the existing ones. Let us discuss some parts of reasons.
>
> - Training models after compression in order to recover accuracy is as important (if not more) as compressing weights. We have found that occasional distortions (not compressing weights for every mini-batch like previous techniques), relatively large learning rate, and training batches in full-precision (unlike previous ones which store compressed weights during entire training) would be the key to recovering or even increasing the accuracy.
> - Exploring large search space in much wider area is suggested in this paper through large distortion step and large learning rate (note that many compression-aware techniques perform compression at every batch has distortion step of “1” while much smaller learning rate for retraining that normal training is chosen). As we discussed in the paper, investigating various local minima is crucial for good model compression.
> - Our pruning method is fundamentally different from the previous ones because we do not incorporate a masking layer. While previous pruning ideas keep zero weights during training, we do not have any zero weights at any moment except at the weight distortion step.
> - Our low-rank approximation is also unique one since 1) we do not alter the structure for training even after performing SVD, 2) very high learning rate associated with transient accuracy loss is allowed for DeepTwist, and 3) we change SV spectrum continuously while the previous ones perform SVD only once (in practice, retraining low-rank approximated model has been considered to be very difficult, if not impossible).
> - Even though our pruning method is even simpler compared to the previous ones, compression rate is significantly better or very close to the one based on sophisticated Bayesian inference model.
> - Low-rank approximation results on PTB (Figure 2) shows even higher compression rate compared with weight pruning (Table 3), which is surprising to us because pruning has been known to show much higher compression ratio compared with SVD (fine-grain vs. coarse-grain or structured).
> - Quantization is performed also in a very different way. Unlike previous ones, we do not consider quatization during
>  training. “Do not perform quantization at every batch, but instead recover accuracy through full-precision training, high learning rate, and occasional quantization” is the key message.
> - Overall, our occasional compression is a significant one since we can greatly reduce amount of computation overhead from compression.
>
> If our technique is a simple extension from the previous ones, we could not obtain such impressive results with high compression rate and improved accuracy. We believe that our paper suggests a wide view on how model compression should be performed.

---

> > ### Comment · AnonReviewer2 · 2018-12-10
> > **If retraining with uncompressed model is the key, then more analyses/results will be great.**
> >
> > Thank you for your responses. Overall, I believe all your arguments above can be summarized as a single point: instead of applying a "hard" compression, it is important to retain full degree of freedom after each distortion step (essentially compress then decompress). However, I still view them as a minor extension to the existing work. Besides, this point is also not clearly motivated in the paper. If this is an important step to get things to work in practice, it will be much more informative and insightful to include analyses and results that clearly show the importance of retraining with uncompressed model.

---

> > > ### Author Response · Authors · 2018-12-10
> > > **This work suggests a new direction in model compression study**
> > >
> > > Thank you for the reply.
> > > We believe that this paper can be a motivation for the model compression community to rethink model compression.
> > >
> > > Recently, increasing number of model compression papers involve not only 'hard' compression for every mini-batch but also more hyper-parameters and ask modifications to the training algorithms to take into account the effects of model compression (most of advanced model compression ideas introduce dedicated feedforward, backward, and/or update steps to make them aware of model compression).
> > > DeepTwist suggests that such efforts need to be re-considered because our experimental results show that compression is necessary only at a distortion step (which is larger than 1).
> > >
> > > Admittedly, it would be a lot better to include thorough analysis on the importance of retraining with uncompressed model. However, we wanted to show such importance empirically first to demonstrate its effectiveness in achieving high compression and good accuracy using various compression techniques. To prove its wide impacts on the existing compression formats, we had to spend good amount of space for experimental results. Even though we tried to explain the motivation in Section 3, we believe it would be challenging in general to derive thorough analyses especially when DeepTwist involves exploring various local minima (as shown in Figure 2), since such analysis is also challenging for large-batch problems and generalization/memorization issues. We will continue our study on the theoretical background of DeepTwist as our future work.

---

### Author Response · Authors · 2018-11-14
**Contributions of this paper (To respond to some reviewers' comments)**

To address some concerns of the reviewers, let us summarize some major contributions of this paper.

1. Introducing distortion step
--> Enables 'Exploration vs. Exploitation' to search for a particular local minimum dedicated to model compression. It also reduces the amount of compression computation overhead significantly. Finding a good distortion step is more important than designing complicated distortion functions.
2. Removing lots of hyper-parameters required to set sophisticated prior information. Our Algorithms 1,2, and 3 have the basic compression format, but they are enough to gain very high compression rates (while recent papers introduce more and more hyper-parameters) which have been only possible with stronger prior information, assumptions, and various heuristic approaches.
3. Moreover, we do not modify underlying training algorithms unlike recent papers with heavy 'compression-aware' ideas along with significantly increased training time, which is a general concern on model compression.
4. As a result, model compression researchers can focus on the fundamental representation format after compression with only basic prior information model on the parameters, without a lot of heuristic and hand-crafted hyper-parameters for each dedicated compression technique (that's why Algorithm 1,2, and 3 can be a lot simpler than other papers due to the concept of 'distortion step' and DeepTwist can be a general framework for model compression)
5. We have shown that our local minimum exploration method can lead to a good regularization effect (much improved accuracy especially for RNN)
6. We could achieve a stable retraining procedure with low-rank approximation (with surprisingly high learning rate)

---

> ### Comment · AnonReviewer1 · 2018-12-10
> **Claims and conclusions are not rigorous.**
>
> I want to thank the authors for the paper and for the rebuttal.
>
> In this current stage of this paper, there are too many claims and conclusions which are not rigorous. I will summarise some of my concerns in the following paragraphs.
>
> The first line of the paper says, “Model compression has been introduced to reduce the required hardware resources while maintaining the model accuracy.” However, there are no further details on how the proposed method can help to reduce the required hardware resources.
>
> From the comments, the introduction of the 'distortion step' is one of the main contributions. The ‘distortion step’ related results are described mostly for the pruning and for the other approaches ‘distortion step’ was set to certain values without rigorous reasoning or experiments. More discussion and experiments with ‘distortion step’ could make the impact clearer.
>
> Another claim is that “Removing lots of hyper-parameters required to set sophisticated prior information”. I don’t see much justification of this claim in the paper. Can the choice of 'distortion step' be considered as an additional hyperparameter?
>
> The claim on regularization is not clear to me. First of all, any technique to reduce the model size plays a role of a regularizer that reduces the testing error. Can’t we get similar error reduction by using different hyperparameter values of the existing methods? The impact of the proposed framework on local minimum exploration and regularization has not been explained properly in the paper.
>
> Overall, this paper needs more work in my opinion.

---

> > ### Author Response · Authors · 2018-12-10
> > **Thank you for the comment**
> >
> > We appreciate your time and efforts for this comment.
> > Please allow us to address your concerns.
> >
> > [Too many claims and conclusions]
> > - Even though we elaborated distinguished points in the paper above compared with the previous approaches, our message is as simple as 'Distortion step and high learning rates can provide better compression rate and/or model accuracy without modifying training algorithms to be aware of model compression.'
> >
> > [Reduced required hardware resources]
> > - We believe that the impacts of pruning, quantization, and low-rank approximations on the required hardware resources have been introduced and discussed in details in many previous papers. For example, quantization reduces memory footprint according to the number of bits to represent weights while low-rank approximation decomposes a large weight matrix into two much smaller matrices (both memory footprint and amount of computations are reduced). Since we do not present a new compression format in the paper (instead, we present how some well-known compression techniques can be significantly enhanced by DeepTwist), we have not discussed how hardware resources can be reduced by pruning and so on.
> >
> > [Distortion step values]
> > - As we wrote in the footnote in the page 4, distortion step shows a low sensitivity to the accuracy for other compression techniques as well. In general, as long as distortion step is not too large (e.g., more than a few epochs) nor too small (e.g., every mini-batch), different distortion steps present similar accuracy as shown in Table 2. We will include experimental results similar to Table 2 for quantization and low-rank approximation in the future.
> >
> > [Removing hyper-parameters]
> > - As we wrote in the introduction, distortion step is the only extra hyper-parameter (which is independent of compression techniques). On the other hand, previous strong compression techniques involve lots of model-specific hyper-parameters. For example, dynamic network surgery introduces thresholds for splicing weights while such thresholds should be separately and empirically obtained for 'each' layer. Sparse VD also involves empirical numbers required for Bayesian statistics. In Table 1, DeepTwist (with distortion step as the only one additional hyper-parameter) shows impressive compression ratio while the results with dynamic network surgery and sparse VD are based on lots of such sophisticated hyper-parameters. Hence, the sentence "Removing lots of ..." is rather summarizing the issues of existing advanced compression techniques.
> >
> > [Regularization effect]
> > Even though it is difficult to visualize local minimum exploration, Figure 2(b) is an indirect way to show how vastly different local minima can be searched by DeepTwist. Table 3 also shows that DeepTwist (with high learning rates) finds different local minima and results in better accuracy even though the pruning rate is the same.
> > In many previous papers, specific numbers for hyper-parameters are provided without justification (while we show the justification in Table 2). Sometimes, even those numbers are not described in the paper (like dynamic network surgery). We borrowed numbers from those papers based on the belief that the authors did their best to obtain the best empirical results.

---

### Meta-Review · Area_Chair1 · 2018-12-14
**Work would be strengthened by additional analyses, and measuring computational resource reduction after applying technique.**

**Confidence:** 4
**Recommendation:** Reject

**Metareview:**

The authors propose a framework for compressing neural network models which involves applying a weight distortion function periodically as part of training. The proposed approach is relatively simple to implement, and is shown to work for weight pruning, low-rank compression and quantization, without sacrificing accuracy.
However, the reviewers had a number of concerns about the work. Broadly, the reviewers felt that the work was incremental. Further, if the proposed techniques are important to get the approach to work well in practice, then the paper would be significantly strengthened by further analyses. Finally, the reviewers noted that the paper does not consider whether the specific weight pruning strategies result in a reduction of computational resources beyond potential storage savings, which would be important if this method is to be used in practice.

Overall, the AC tends to agree with the reviewers criticisms. The authors are encouraged to address some of these issues in future revisions of the work.